# Detecting Overfitting via Adversarial Examples

**Roman Werpachowski**      **András György**      **Csaba Szepesvári**
DeepMind, London, UK
{romanw,agyorgy,szepi}@google.com

## Abstract

The frequent reuse of test sets in popular benchmark problems raises doubts about the credibility of reported test-error rates. Verifying whether a learned model is overfitted to a test set is challenging as independent test sets drawn from the same data distribution are usually unavailable, while other test sets may introduce a distribution shift. We propose a new hypothesis test that uses only the original test data to detect overfitting. It utilizes a new unbiased error estimate that is based on adversarial examples generated from the test data and importance weighting. Overfitting is detected if this error estimate is sufficiently different from the original test error rate. We develop a specialized variant of our test for multiclass image classification, and apply it to testing overfitting of recent models to the popular ImageNet benchmark. Our method correctly indicates overfitting of the trained model to the training set, but is not able to detect any overfitting to the test set, in line with other recent work on this topic.

## 1   Introduction

Deep neural networks achieve impressive performance on many important machine learning benchmarks, such as image classification [18, 19, 28, 27, 16], automated translation [2, 31] or speech recognition [9, 15]. However, the benchmark datasets are used a multitude of times by researchers worldwide. Since state-of-the-art methods are selected and published based on their performance on the corresponding test set, it is typical to see results that continuously improve over time; see, e.g., the discussion of Recht et al. [25] and Figure 1 for the performance improvement of classifiers published for the popular CIFAR-10 image classification benchmark [18].

This process may naturally lead to models overfitted to the test set, rendering test error rate (the average error measured on the test set) an unreliable indicator of the actual performance. Detecting whether a model is overfitted to the test set is challenging, since independent test sets drawn from the same data distribution are generally not available, while alternative test sets often introduce a distribution shift.

To estimate the performance of a model on unseen data, one may use generalization bounds to get upper bounds on the expected error rate. The generalization bounds are also applicable when the model and the data are dependent (e.g., for cross validation or for error estimates based

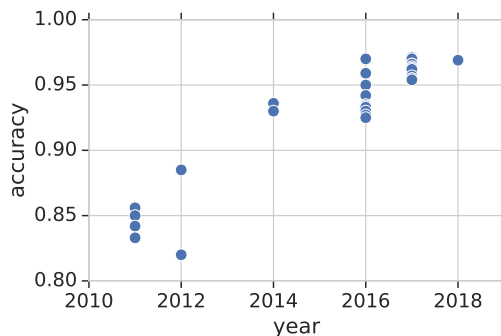

Figure 1: Accuracy of image classifiers on the CIFAR-10 test set, by year of publication (data from [25]).

on the training data or the reused test data), but they usually lead to loose error bounds. Therefore, although much tighter bounds are available if the test data and the model are independent, comparing

confidence intervals constructed around the training and test error rates leads to an underpowered test for detecting the dependence of a model on the test set. Recently, several methods have been proposed that allow the reuse of the test set while keeping the validity of test error rates [10]. However, these are *intrusive*: they require the user to follow a strict protocol of interacting with the test set and are thus not applicable in the more common situation when enforcing such a protocol is impossible.

In this paper we take a new approach to the challenge of detecting overfitting of a model to the test set, and devise a *non-intrusive* statistical test that does not restrict the training procedure and is based on the original test data. To this end, we introduce a new error estimator that is less sensitive to overfitting to the data; our test rejects the independence of the model and the test data if the new error estimate and the original test error rate are too different. The core novel idea is that the new estimator is based on adversarial examples [14], that is, on data points[1] that are not sampled from the data distribution, but instead are cleverly crafted based on existing data points so that the model errs on them. Several authors showed that the best models learned for the above-mentioned benchmark problems are highly sensitive to adversarial attacks [14, 23, 30, 6, 7, 24]: for instance, one can often create adversarial versions of images properly classified by a state-of-the-art model such that the model will misclassify them, yet the adversarial perturbations are (almost) undetectable for a human observer; see, e.g., Figure 2, where the adversarial image is obtained from the original one by a carefully selected translation.

The *adversarial (error) estimator* proposed in this work uses adversarial examples (generated from the test set) together with importance weighting to take into account the change in the data distribution (covariate shift) due to the adversarial transformation. The estimator is un-biased and has a smaller variance than the standard test error rate if the test set and the model are independent.[2] More importantly, since it is based on adversarially generated data points, the adversarial estimator is expected to differ significantly from the test error rate if the model is overfitted to the test set, providing a way to detect test set overfitting. Thus, the test error rate and the adversarial error estimate (calcu-

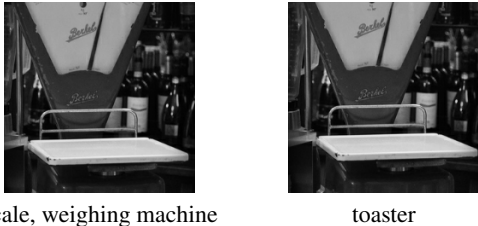

scale, weighing machine          toaster

Figure 2: Adversarial example for the ImageNet dataset generated by a $(5, -5)$ translation: the original example (left) is correctly classified by the VGG16 model [27] as "scale, weighing machine," the adversarially generated example (right) is classified as "toaster," while the image class is the same for any human observer.

lated based on the same test set) must be close if the test set and the model are independent, and are expected to be different in the opposite case. In particular, if the gap between the two error estimates is large, the independence hypothesis (i.e., that the model and the test set are independent) is dubious and will be rejected. Combining results from multiple training runs, we develop another method to test overfitting of a model architecture and training procedure (for simplicity, throughout the paper we refer to both together as the *model architecture*). The most challenging aspect of our method is to construct adversarial perturbations for which we can calculate importance weights, while keeping enough degrees of freedom in the way the adversarial perturbations are generated to maximize power, the ability of the test to detect dependence when it is present.

To understand the behavior of our tests better, we first use them on a synthetic binary classification problem, where the tests are able to successfully identify the cases where overfitting is present. Then we apply our independence tests to state-of-the-art classification methods for the popular image classification benchmark, ImageNet [8]. As a sanity check, in all cases examined, our test rejects (at confidence levels close to 1) the independence of the individual models from their respective training sets. Applying our method to VGG16 [27] and Resnet50 [16] models/architectures, their *independence to the ImageNet test set cannot be rejected at any reasonable confidence*. This is in agreement with recent findings of [26], and provides additional evidence that despite of the existing danger, it is likely that no overfitting has happened during the development of ImageNet classifiers.

The rest of the paper is organized as follows: In Section 2, we introduce a formal model for error estimation using adversarial examples, including the definition of adversarial example generators.

The new overfitting-detection tests are derived in Section 3, and applied to a synthetic problem in Section 4, and to the ImageNet image classification benchmark in Section 5. Due to space limitations, some auxiliary results, including the in-depth analysis of our method on the synthetic problem, are relegated to the appendix.

## 2   Adversarial Risk Estimation

We consider a classification problem with deterministic (noise-free) labels, which is a reasonable assumption for many practical problems, such as image recognition (we leave the extension of our method to noisy labels for future work). Let $\mathcal{X} \subset \mathbb{R}^D$ denote the input space and $\mathcal{Y} = \{0, \ldots, K-1\}$ the set of labels. Data is sampled from the distribution $\mathcal{P}$ over $\mathcal{X}$, and the class label is determined by the *ground truth* function $f^* : \mathcal{X} \to \mathcal{Y}$. We denote a random vector drawn from $\mathcal{P}$ by $X$, and its corresponding class label by $Y = f^*(X)$. We consider deterministic classifiers $f : \mathcal{X} \to \mathcal{Y}$. The performance of $f$ is measured by the zero-one loss: $L(f, x) = \mathbb{I}(f(x) \neq f^*(x))$,[3] and the *expected error* (also known as the *risk* or *expected risk* in the learning theory literature) of the classifier $f$ is defined as $R(f) = \mathbb{E}[\mathbb{I}(f(X) \neq Y)] = \int_{\mathcal{X}} L(f, x) \mathrm{d}\mathcal{P}(x)$.

Consider a test dataset $S = \{(X_1, Y_1) \ldots, (X_m, Y_m)\}$ where the $X_i$ are drawn from $\mathcal{P}$ independently of each other and $Y_i = f^*(X_i)$. In the learning setting, the classifier $f$ usually also depends on some randomly drawn training data, hence is random itself. If $f$ is (statistically) independent from $S$, then $L(f, X_1), \ldots, L(f, X_m)$ are i.i.d., thus the empirical error rate

$$\widehat{R}_S(f) = \frac{1}{m} \sum_{i=1}^{m} L(f, X_i) = \frac{1}{m} \sum_{i=1}^{m} \mathbb{I}(f(X_i) \neq Y_i)$$

is an unbiased estimate of $R(f)$ for all $f$; that is, $R(f) = \mathbb{E}[\widehat{R}_S(f)|f]$. If $f$ and $S$ are not independent, the performance guarantees on the empirical estimates available in the independent case are significantly weakened; for example, in case of overfitting to $S$, the empirical error rate is likely to be much smaller than the expected error.

Another well-known way to estimate $R(f)$ is to use *importance sampling* (IS) [17]: instead of sampling from the distribution $\mathcal{P}$, we sample from another distribution $\mathcal{P}'$ and correct the estimate by appropriate reweighting. Assuming $\mathcal{P}$ is absolutely continuous with respect to $\mathcal{P}'$ on the set $E = \{x \in \mathcal{X} : L(f, x) \neq 0\}$, $R(f) = \int_{\mathcal{X}} L(f, x) \mathrm{d}\mathcal{P}(x) = \int_E L(f, x) h(x) \mathrm{d}\mathcal{P}'(x)$, where $h = \frac{\mathrm{d}\mathcal{P}}{\mathrm{d}\mathcal{P}'}$ is the density (Radon-Nikodym derivative) of $\mathcal{P}$ with respect to $\mathcal{P}'$ on $E$ ($h$ can be defined to have arbitrary finite values on $\mathcal{X} \setminus E$). It is well known that the the corresponding empirical error estimator

$$\widehat{R}'_{S'}(f) = \frac{1}{m} \sum_{i=1}^{m} L(f, X'_i) h(X'_i) = \frac{1}{m} \sum_{i=1}^{m} \mathbb{I}(f(X'_i) \neq Y'_i) h(X'_i) \tag{1}$$

obtained from a sample $S' = \{(X'_1, Y'_1), \ldots, (X'_m, Y'_m)\}$ drawn independently from $\mathcal{P}'$ is unbiased (i.e., $\mathbb{E}[\widehat{R}'_{S'}(f)|f] = R(f)$) if $f$ and $S'$ are independent.

The variance of $\widehat{R}'_{S'}$ is minimized if $\mathcal{P}'$ is the so-called zero-variance IS distribution, which is supported on $E$ with $h(x) = \frac{R(f)}{L(f,x)}$ for all $x \in E$ (see, e.g., [4, Section 4.2]). This suggest that an effective sampling distribution $\mathcal{P}'$ should concentrate on points where $f$ makes mistakes, which also facilitates that $\widehat{R}'_{S'}(f)$ become large if $f$ is overfitted to $S$ and hence $\widehat{R}_S(f)$ is small. We achieve this through the application of adversarial examples.

### 2.1   Generating adversarial examples

In this section we introduce a formal framework for generating adversarial examples. Given a classification problem with data distribution $\mathcal{P}$ and ground truth $f^*$, an *adversarial example generator* (AEG) for a classifier $f$ is a (measurable) mapping $g : \mathcal{X} \to \mathcal{X}$ such that

- (G1) $g$ preserves the class labels of the samples, that is, $f^*(x) = f^*(g(x))$ for $\mathcal{P}$-almost all $x$;
- (G2) $g$ does not change points that are incorrectly classified by $f$, that is, $g(x) = x$ if $f(x) \neq f^*(x)$ for $\mathcal{P}$-almost all $x$.

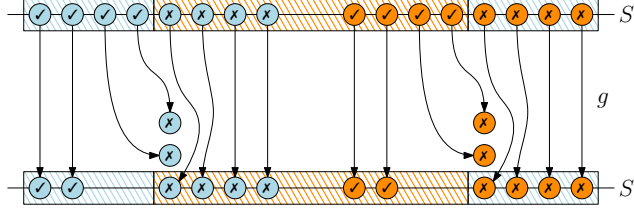

Figure 3: Generating adversarial examples. The top row depicts the original dataset $S$, with blue and orange points representing the two classes. The classifier's prediction is represented by the color of the striped areas (checkmarks and crosses denote if a point is correctly or incorrectly classified). The arrows show the adversarial transformations via the AEG $g$, resulting in the new dataset $S'$; misclassified points are unchanged, while some correctly classified points are moved, but their original class label is unchanged. If the original data distribution is uniform over $S$, the transformation $g$ is density preserving, but not measure preserving: after the transformation the two rightmost correctly classified points in each class have probability 0, while the leftmost misclassified point in each class has probability $3/16$; hence, the density $h_g$ for the latter points is $1/3$.

Figure 3 illustrates how an AEG works. In the literature, an adversarial example $g(x)$ is usually generated by staying in a small vicinity of the original data point $x$ (with respect to, e.g., the 2- or the max-norm) and assuming that the resulting label of $g(x)$ is the same as that of $x$ (see, e.g., [14, 6]). This foundational assumption—which is in fact a margin condition on the distribution—is captured in condition (G1). (G2) formalizes the fact that there is no need to change samples which are already misclassified. Indeed, existing AEGs comply with this condition.

The performance of an AEG is usually measured by how successfully it generates misclassified examples. Accordingly, we call a point $g(x)$ a *successful adversarial example* if $x$ is correctly classified by $f$ and $f(g(x)) \neq f(x)$ (i.e., $L(f, x) = 0$ and $L(f, g(x)) = 1$).

In the development of our AEGs for image recognition tasks, we will make use of another condition. For simplicity, we formulate this condition for distributions $\mathcal{P}$ that have a density $\rho$ with respect to the uniform measure on $\mathcal{X}$, which is assumed to exist (notable cases are when $\mathcal{X}$ is finite, or $\mathcal{X} = [0, 1]^D$ or when $\mathcal{X} = \mathbb{R}^D$; in the latter two cases the uniform measure is the Lebesgue measure). The assumption states that the AEG needs to be *density-preserving*:

(G3)  $\rho(x) = \rho(g(x))$ for $\mathcal{P}$-almost all $x$.

Note that a density-preserving map may not be measure-preserving (the latter means that for all measurable $A \subset \mathcal{X}$, $\mathcal{P}(A) = \mathcal{P}(g(A))$).

We expect (G3) to hold when $g$ perturbs its input by a small amount and if $\rho$ is sufficiently smooth. The assumption is reasonable for, e.g., image recognition problems (at least in a relaxed form, $\rho(x) \approx \rho(g(x))$) where we expect that very close images will have a similar likelihood as measured by $\rho$. An AEG employing image translations, which satisfies (G3), will be introduced in Section 5. Both (G1) and (G3) can be relaxed (to a soft margin condition or allowing a slight change in $\rho$, resp.) at the price of an extra error term in the analysis that follows.

For a fixed AEG $g : \mathcal{X} \to \mathcal{X}$, let $\mathcal{P}_g$ be the distribution of $g(X)$ where $X \sim \mathcal{P}$ ($\mathcal{P}_g$ is known as the pushforward measure of $\mathcal{P}$ under $g$). Further, let $h_g = \frac{d\mathcal{P}}{d\mathcal{P}_g}$ on $E = \{x \ : \ L(f, x) \neq 0\}$ and arbitrary otherwise. It is easy to see that, on $E$, $h_g(x)$ is well-defined and $h_g \leq 1$. For any measurable $A \subset E$

$$\mathcal{P}_g(A) = \mathbb{P}(g(X) \in A) \geq \mathbb{P}(g(X) \in A, X \in E) = \mathbb{P}(X \in A) = \mathcal{P}(A)$$

where the second to last equality holds because $g(X) = X$ for any $X \in E$ under condition (G2). Thus, $\mathcal{P}(A) \leq \mathcal{P}_g(A)$ for any measurable $A \subset E$, which implies that $h_g$ is well-defined on $E$ and $h_g(x) \leq 1$ for all $x \in E$.

One may think that (G3) implies that $h_g(x) = 1$ for all $x \in E$. However, this does not hold. For example, if $\mathcal{P}$ is a uniform distribution, any $g : \mathcal{X} \to \operatorname{supp} \mathcal{P}$ satisfies (G3), where $\operatorname{supp} \mathcal{P} \subset \mathcal{X}$ denotes the support of the distribution $\mathcal{P}$. This is also illustrated in Figure 3.

## 2.2 Risk estimation via adversarial examples

Combining the ideas of this section so far, we now introduce unbiased risk estimates based on adversarial examples. Our goal is to estimate the error-rate of $f$ through an adversarially generated

sample $S' = \{(X_1', Y_1), \ldots, (X_m', Y_m)\}$ obtained through an AEG $g$, where $X_i' = g(X_i)$ with $X_1, \ldots, X_m$ drawn independently from $\mathcal{P}$ and $Y_i = f^*(X_i)$. Since $g$ satisfies (G1) by definition, the original example $X_i$ and the corresponding adversarial example $X_i'$ have the same label $Y_i$. Recalling that $h_g = \mathrm{d}\mathcal{P}/\mathrm{d}\mathcal{P}_g \leq 1$ on $E = \{x \in \mathcal{X} : L(f, x) = 1\}$, one can easily show that the importance weighted adversarial estimate

$$\widehat{R}_g(f) = \frac{1}{m} \sum_{i=1}^{m} \mathbb{I}(f(X_i') \neq Y_i) h_g(X_i') \tag{2}$$

obtained from (1) for the adversarial sample $S'$ has smaller variance than that of the empirical average $\widehat{R}_S(f)$, while both are unbiased estimates of $R(f)$. Recall that both $\widehat{R}_g(f)$ and $\widehat{R}_S(f)$ are unbiased estimates of $R(f)$ with expectation $\mathbb{E}[\widehat{R}_g(f)] = \mathbb{E}[\widehat{R}_S(f)] = R(f)$, and so

$$\begin{aligned}
\mathbb{V}[\widehat{R}_g(f)] &= \frac{1}{m} \left( \mathbb{E}[L(f, g(X))^2 h_g(g(X))^2] - R(f)^2 \right) \\
&\leq \frac{1}{m} \left( \mathbb{E}[L(f, g(X)) h_g(g(X))] - R^2(f) \right) = \frac{1}{m} \left( R(f) - R^2(f) \right) = \mathbb{V}[\widehat{R}_S(f)] .
\end{aligned}$$

Intuitively, the more successful the AEG is (i.e., the more classification error it induces), the smaller the variance of the estimate $\widehat{R}_g(f)$ becomes.

## 3 Detecting overfitting

In this section we show how the risk estimates introduced in the previous section can be used to test the *independence hypothesis* that

(H) the sample $S$ and the model $f$ are independent.

If (H) holds, $\mathbb{E}[\widehat{R}_g(f)] = \mathbb{E}[\widehat{R}_S(f)] = R(f)$, and so the difference $T_{S,g}(f) = \widehat{R}_g(f) - \widehat{R}_S(f)$ is expected to be small. On the other hand, if $f$ is overfitted to the dataset $S$ (in which case $\widehat{R}_S(f) < R(f)$), we expect $\widehat{R}_S(f)$ and $\widehat{R}_g(f)$ to behave differently (the latter being less sensitive to overfitting) since (i) $\widehat{R}_g(f)$ depends also on examples previously unseen by the training procedure; (ii) the adversarial transformation $g$ aims to increase the loss, countering the effect of overfitting; (iii) especially in high dimensional settings, in case of overfitting one may expect that there are misclassified points very close to the decision boundary of $f$ which can be found by a carefully designed AEG. Therefore, intuitively, (H) can be rejected if $|T_{S,g}(f)|$ exceeds some appropriate threshold.

### 3.1 Test based on confidence intervals

The simplest way to determine the threshold is based on constructing confidence intervals for these estimator based on concentration inequalities. Under (H), standard concentration inequalities, such as the Chernoff or empirical Bernstein bounds [3], can be used to quantify how fast $\widehat{R}_S$ and $\widehat{R}_g(f)$ concentrate around the expected error $R(f)$. In particular, we use the following empirical Bernstein bound [22]: Let $\bar{\sigma}_S^2 = (1/m) \sum_{i=1}^{m} (L(f, X_i) - \widehat{R}_S(f))^2$ and $\bar{\sigma}_g^2 = (1/m) \sum_{i=1}^{m} (L(f, g(X_i)) h_g(g(X_i)) - \widehat{R}_g(f))^2$ denote the empirical variance of $L(f, X_i)$ and $L(f, g(X_i)) h_g(g(X_i))$, respectively. Then, for any $0 < \delta \leq 1$, with probability at least $1 - \delta$,

$$|\widehat{R}_S(f) - R(f)| \leq B(m, \bar{\sigma}_S^2, \delta, 1), \tag{3}$$

where $B(m, \sigma^2, \delta, 1) = \sqrt{\frac{2\sigma^2 \ln(3/\delta)}{m}} + \frac{3 \ln(3/\delta)}{m}$ and we used the fact that the range of $L(f, x)$ is 1 (the last parameter of $B$ is the range of the random variables considered). Similarly, with probability at least $1 - \delta$,

$$|\widehat{R}_g(f) - R(f)| \leq B(m, \bar{\sigma}_g^2, \delta, 1). \tag{4}$$

It follows trivially from the union bound that if the independence hypothesis (H) holds, the above two confidence intervals $[\widehat{R}_S(f) - B(m, \bar{\sigma}_S^2, \delta, 1), \widehat{R}_S(f) + B(m, \bar{\sigma}_S^2, \delta, 1)]$ and $[\widehat{R}_g(f) -$

$B(m, \bar{\sigma}_g^2, \delta, 1), \widehat{R}_S(f) + B(m, \bar{\sigma}_g^2, \delta, 1)]$, which both contain $R(f)$ with probability at least $1 - \delta$, intersect with probability at least $1 - 2\delta$.

On the other hand, if $f$ and $S$ are not independent, the performance guarantees (3) and (4) may be violated and the confidence intervals may become disjoint. If this is detected, we can reject the independence hypothesis (H) at a confidence level $1 - 2\delta$ or, equivalently, with $p$-value $2\delta$. In other words, we reject (H) if the absolute value of the difference of the estimates $T_{S,g}(f) = \widehat{R}_g(f) - \widehat{R}_S(f)$ exceeds the threshold $B(m, \bar{\sigma}_S^2, \delta, 1) + B(m, \bar{\sigma}_g^2, \delta, 1)$ (note that $\mathbb{E}[T_{S,g}(f)] = 0$ if $S$ and $f$ are independent).

## 3.2 Pairwise test

A smaller threshold for $|T_{S,g}(f)|$, and hence a more effective independence test, can be devised if instead of independently estimating the behavior of $\widehat{R}_S$ and $\widehat{R}_g(f)$, one utilizes their apparent correlation. Indeed, $T_{S,g}(f) = (1/m) \sum_{i=1}^m T_{i,g}(f)$ where

$$T_{i,g}(f) = L(f, g(X_i)) h_g(g(X_i)) - L(f, X_i) \tag{5}$$

and the two terms in $T_{i,g}(f)$ have the same mean and are typically highly correlated by the construction of $g$. Thus, we can apply the empirical Bernstein bound [22] to the pairwise differences $T_{i,g}(f)$ to set a tighter threshold in the test: if the independence hypothesis (H) holds (i.e., $S$ and $f$ are independent), then for any $0 < \delta < 1$, with probability at least $1 - \delta$,

$$|T_{S,g}(f)| \leq B(m, \bar{\sigma}_T^2, \delta, U) \tag{6}$$

with $B(m, \sigma^2, \delta, U) = \sqrt{\frac{2\sigma^2 \ln(3/\delta)}{m}} + \frac{3U \ln(3/\delta)}{m}$, where $\bar{\sigma}_T^2 = (1/m) \sum_{i=1}^m (T_i(f) - T_{S,g}(f))^2$ is the empirical variance of the $T_{i,g}(f)$ terms and $U = \sup T_{i,g}(f) - \inf T_{i,g}(f)$; we also used the fact that the expectation of each $T_{i,g}(f)$, and hence that of $T_{S,g}(f)$, is zero. Since $h_g \leq 1$ if $L(f,x) = 1$ (as discussed in Section 2.2), it follows that $U \leq 2$, but further assumptions (such as $g$ being density preserving) can result in tighter bounds.

This leads to our pairwise dependence detection method:

*if $|T_{S,g}(f)| > B(m, \bar{\sigma}_T^2, \delta, 2)$, reject* (H) *at a confidence level $1 - \delta$ (p-value $\delta$).*

For a given statistic $(|T_{S,g}(f)|, \bar{\sigma}_T^2)$, the largest confidence level (smallest $p$-value) at which (H) can be rejected can be calculated by setting the value of the statistic $|T_{S,g}(f)| - B(m, \bar{\sigma}_T^2, \delta, 2)$ to zero and solving for $\delta$. This leads to the following formula for the $p$-value (if the solution is larger than 1, which happens when the bound (6) is loose, $\delta$ is capped at 1):

$$\delta = \min \left\{ 1, 3e^{-\frac{m}{9U^2} \left( \bar{\sigma}_T^2 + 3U|T_{S,g}(f)| - \bar{\sigma}_T \sqrt{\bar{\sigma}_T^2 + 6U|T_{S,g}(f)|} \right)} \right\}. \tag{7}$$

Note that in order for the test to work well, we not only need the test statistic $T_{S,g}(f)$ to have a small variance in case of independence (this could be achieved if $g$ were the identity), but we also need the estimators $\widehat{R}_S(f)$ and $\widehat{R}_g(f)$ behave sufficiently differently if the independence assumption is violated. The latter behavior is encouraged by stronger AEGs, as we will show empirically in Section 5.2 (see Figure 5 in particular).

## 3.3 Dependence detector for randomized training

The dependence between the model and the test set can arise from (i) selecting the "best" random seed in order to improve the test set performance and/or (ii) tweaking the model architecture (e.g., neural network structure) and hyperparameters (e.g., learning-rate schedule). If one has access to a single instance of a trained model, these two sources cannot be disentangled. However, if the model architecture and training procedure is fully specified and computational resources are adequate, it is possible to isolate (i) and (ii) by retraining the model multiple times and calculating the $p$-value for every training run separately. Assuming $N$ models, let $f_j, j = 1, \ldots, N$ denote the $j$-th trained model and $p_j$ the $p$-value calculated using the pairwise independence test (6) (i.e., from Eq. 7 in Section 3). We can investigate the degree to which (i) occurs by comparing the $p_j$ values with the corresponding test set error rates $R_S(f_j)$. To investigate whether (ii) occurs, we can average over the randomness of the training runs.

For every example $X_i \in S$, consider the average test statistic $\bar{T}_i = \frac{1}{N} \sum_{j=1}^{N} T_{i,g_j}(f_j)$, where $T_{i,g_j}(f_j)$ is the statistic (5) calculated for example $X_i$ and model $f_j$ with AEG $g_j$ selected for model $f_j$ (note that AEGs are model-dependent by construction). If, for each $i$ and $j$, the random variables $T_i(f_j)$ are independent, then so are the $\bar{T}_i$ (for all $i$). Hence, we can apply the pairwise dependence detector (6) with $\bar{T}_i$ instead of $T_i$, using the average $\bar{T}_S = (1/m) \sum_{i=1}^{m} \bar{T}_i$ with empirical variance $\bar{\sigma}_{T,N}^2 = (1/m) \sum_{i=1}^{m} (\bar{T}_i - \bar{T}_S)^2$, giving a single $p$-value $p_N$. If the training runs vary enough in their outcomes, different models $f_j$ err on different data points $X_j$, leading to $\bar{\sigma}_{T,N}^2 < \bar{\sigma}_T^2$, and therefore strengthening the power of the dependence detector. For brevity, we call this independence test an $N$-model test.

## 4    Synthetic experiments

First we verify the effectiveness of our method on a simple linear classification problem. Due to space limitations, we only convey high-level results here, details are given in Appendix A. We assume that the data is linearly separable with a margin and the density $\rho$ is known. We consider a linear classifiers of the form $f(x) = \mathrm{sgn}(w^\top x + b)$ trained with the cross-entropy loss $c$, and we employ a one-step gradient method (which is an $L_2$ version of the fast gradient-sign method of [14, 23]) to define our AEG $g$, which tries to modify a correctly classified point $x$ with label $y$ in the direction of the gradient of the cost function, yielding $x' = x - \varepsilon y w / \|w\|_2$, where $\varepsilon \geq 0$ is the strength of the attack. To comply with the requirements

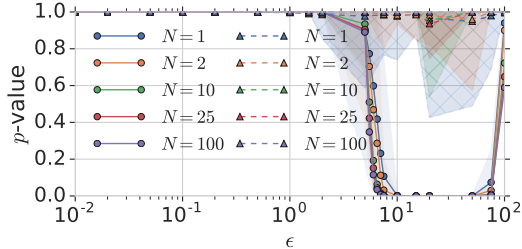

Figure 4: Average $p$-values produced by the independence test in a separable linear classification problem for the cases of both when the model is independent of (dashed lines) and, resp., dependent on (solid lines) the test set.

for an AEG, we define $g$ as follows: $g(x) = x'$ if $L(f, x) = 0$ and $f^*(x) = f^*(x')$ (corresponding to (G2) and (G1), respectively), while $g(x) = x$ otherwise. Therefore, if $x'$ is misclassified by $f$, $x$ and $x'$ are the only points mapped to $x'$ by $g$. This simple form of $g$ and the knowledge of $\rho$ allows to compute the density $h_g$, making it easy to compute the adversarial error estimate (2). Figure 4 shows the average $p$-values produced by our $N$-model independence test for a dependent (solid lines) and an independent (dashed lines) test set. It can be seen that in the dependent case the test can reject independence with high confidence for a large range of attack strength $\varepsilon$, while the independence hypothesis is not rejected in the case of true independence. More details (including why only a range of $\varepsilon$ is suitable for detecting overfitting) are given in Appendix A.

## 5    Testing overfitting on ImageNet

In the previous section we showed that the proposed adversarial-example-based dependence test works for a synthetic problem where the densities can be computed exactly. In this section we apply our estimates to a popular image classification benchmark, ImageNet [8]; here the main issue is to find sufficiently strong AEGs that make computing the corresponding densities possible.

To facilitate the computation of the density $h_g$, we only consider density-preserving AEGs as defined by (G3) (recall that (G3) is different from requiring $h_g = 1$). Since in (2) and (5), $h_g(x)$ is multiplied by $L(f, x)$, we only need to determine the density $h_g$ for data points that are misclassified by $f$.

### 5.1    AEGs based on translations

To satisfy (G3), we implement the AEG using translations of images, which have recently been proposed as means of generating adversarial examples [1]. Although relatively weak, such attacks fit our needs well: unless the images are procedurally centered, it is reasonable to assume that translating them by a few pixels does not change their likelihood.[4] We also make the natural assumption that the small translations used do not change the true class of an image. Under these assumptions,

translations by a few pixels satisfy conditions (G1) and (G3). An image-translating function $g$ is a valid AEG if it leaves all misclassified images in place (to comply with (G2)), and either leaves a correctly classified image unchanged or applies a small translation.

The main benefit of using a translational AEG $g$ (with bounded translations) is that its density $h_g(x)$ for an image $x$ can be calculated exactly by considering the set of images $x'$ that can be mapped to $x$ by $g$ (this is due to our assumption (G3)). We considered multiple ways for constructing translational AEGs. The best version (selected based on initial evaluations on the ImageNet training set), which we called the *strongest perturbation*, seeks a non-identical neighbor of a correctly classified image $x$ (neighboring images are the ones that are accessible through small translations) that causes the classifier to make an error with the largest confidence.

Formally, we model images as 3D tensors in $[0,1]^{W \times H \times C}$ space, where $C = 3$ for RGB data, and $W$ and $H$ are the width and height of the images, respectively. Let $\tau_v(x)$ denote the translation of an image $x$ by $v \in \mathbb{Z}^2$ pixels in the (X, Y) plane (here $\mathbb{Z}$ denotes the set of integers). To control the amount of change, we limit the magnitude of translations and allow $v \in \mathcal{V}_\varepsilon = \{u \in \mathbb{Z}^2 : u \neq (0,0), \|u\|_\infty \leq \varepsilon\}$ only, for some fixed positive $\varepsilon$. Thus, we considers AEGs in the form $g(x) \in \{\tau_v(x) : v \in \mathcal{V}\} \cup \{x\}$ if $f(x) = f^*(x)$ and $g(x) = x$ otherwise (if $x$ is correctly classified, we attempt to translate it to find an adversarial example in $\{\tau_v(x) : v \in \mathcal{V}\}$ which is misclassified by $f$, but $x$ is left unchanged if no such point exists). Denoting the density of the pushforward measure $\mathcal{P}_g$ by $\rho_g$, for any misclassified point $x$,

$$\rho_g(x) = \rho(x) + \sum_{v \in \mathcal{V}} \rho(\tau_{-v}(x)) \mathbb{I}(g(\tau_{-v}(x)) = x) = \rho(x) \left( 1 + \sum_{v \in \mathcal{V}} \mathbb{I}(g(\tau_{-v}(x)) = x) \right)$$

where the second equality follows from (G3). Therefore, the corresponding density is

$$h_g(x) = 1/(1 + n(x)) \tag{8}$$

where $n(x) = \sum_{v \in \mathcal{V}} \mathbb{I}(g(\tau_{-v}(x)) = x)$ is the number of neighboring images which are mapped to $x$ by $g$. Note that given $f$ and $g$, $n(x)$ can be easily calculated by checking all possible translations of $x$ by $-v$ for $v \in \mathcal{V}$. It is easy to extend the above to non-deterministic perturbations, defined as distributions over AEGs, by replacing the indicator with its expectation $\mathbb{P}(g(\tau_{-v}(x)) = x | x, v)$ with respect to the randomness of $g$, yielding

$$h_g(x) = \frac{1}{1 + \sum_{v \in \mathcal{V}} \mathbb{P}(g(\tau_{-v}(x)) = x | x, v)} . \tag{9}$$

If $g$ is deterministic, we have $h_g(x) \leq 1/2$ for any successful adversarial example $x$. Hence, for such $g$, the range $U$ of the random variables $T_i$ defined in (5) has a tighter upper bound of 3/2 instead 2 (as $T_i \in [-1, 1/2]$), leading to a tighter bound in (6) and a stronger pairwise independence test. In the experiments, we use this stronger test. We provide additional details about the translational AEGs used in Appendix B.

## 5.2 Tests of ImageNet models

We applied our test to check if state-of-the-art classifiers for the ImageNet dataset [8] have been overfitted to the test set. In particular, we use the VGG16 classifier of [27] and the Resnet50 classifier of [16]. Due to computational considerations, we only analyzed a single trained VGG16 model, while the Resnet50 model was retrained 120 times. The models were trained using the parameters recommended by their respective authors.

The preprocessing procedure of both architectures involves rescaling every image so that the smaller of width and height is 256 and next cropping centrally to size $224 \times 224$. This means that translating the image by $v$ can be trivially implemented by shifting the cropping window by $-v$ without any loss of information for $\|v\|_\infty \leq 16$, because we have enough extra pixels outside the original, centrally located cropping window. This implies that we can compute the densities of the translational AEGs for any $\|v\|_\infty \leq \varepsilon = \lfloor 16/3 \rfloor = 5$ (see Appendix B.1 for detailed explanation). Because the ImageNet data collection procedure did not impose any strict requirements on centering the images [8], it is reasonable to assume (as we do) that small (lossless) translations respect the density-preserving condition (G3).

In our first experiment, we applied our pairwise independence test (6) with the AEGs described in Appendix B (strongest, nearest, and the two random baselines) to all 1,271,167 training examples, as

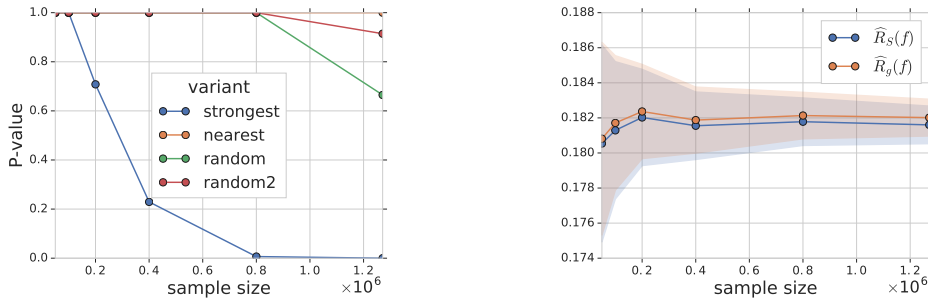

Figure 5: $p$-values for the independence test on the ImageNet training set for different sample sizes and AEG variants (left); original and adversarial risk estimates, $\widehat{R}_S(f)$ and $\widehat{R}_g(f)$, on the ImageNet training set with 97.5% two-sided confidence intervals for the 'strongest attack' AEG (right).

well as to a number of its randomly selected (uniformly without replacement) subsets of different sizes. Besides this being a sanity check, we also used this experiment to select from different AEGs and compare the performance of the pairwise independence test (6) to the basic version of the test described in Section 3.1.

The left graph in Figure 5 shows that with the "strongest perturbation", we were able to reject independence of the trained model and the training samples at a confidence level very close to 1 when enough training samples are considered (to be precise, for the whole training set the confidence level is 99.9994%). Note, however, that the much weaker "smallest perturbation" AEG, as well as the random transformations, are not able to detect the presence of overfitting. At the same time, the graph on the right hand side shows the relative strength of the pairwise independence test compared to the basic version based on independent confidence interval estimates as described in detail in Section 3.1: the 97.5%-confidence intervals of the error estimates $\widehat{R}_S(f)$ and $\widehat{R}_g(f)$ overlap, not allowing to reject independence at a confidence level of 95% (note that here $S$ denotes the training set).

On the other hand, when applied to the test set, we obtained a $p$-value of 0.96, not allowing at all to reject the independence of the trained model and the test set. This result could be explained by the test being too weak, as no overfitting is detected to the *training* set at similar sample sizes (see Figure 5), or simply the lack of overfitting. Similar results were obtained for Resnet50, where even the $N$-model test with $N = 120$ independently trained models resulted a $p$ value of 1, not allowing to reject independence at any confidence level. The view of no overfitting can be backed up in at least two ways: first, "manual" overfitting to the relatively *large* ImageNet test set is hard. Second, since training an ImageNet model was just too computationally expensive until quite recently, only a relatively small number of different architectures were developed for this problem, and the evolution of their design was often driven by computational efficiency on the available hardware. On the other hand, it is also possible that increasing $N$ sufficiently might show evidence of overfitting (this is left for future work).

# 6 Conclusions

We presented a method for detecting overfitting of models to datasets. It relies on an importance-weighted risk estimate from a new dataset obtained by generating adversarial examples from the original data points. We applied our method to the popular ImageNet image classification task. For this purpose, we developed a specialized variant of our method for image classification that uses adversarial translations, providing arguments for its correctness. Luckily, and in agreement with other recent work on this topic [25, 26, 13, 21, 32], we found no evidence of overfitting of state-of-the-art classifiers to the ImageNet test set.

The most challenging aspect of our methods is to construct adversarial perturbations for which we can calculate the importance weights; finding stronger perturbations than the ones based on translations for image classification is an important question for the future. Another interesting research direction is to consider extensions beyond image classification, for example, by building on recent adversarial attacks for speech-to-text methods [5], machine translation [11] or text classification [12].

## Acknowledgements

We thank J. Uesato for useful discussions and advice about adversarial attack methods and sharing their implementations [30] with us, as well as M. Rosca and S. Gowal for help with retraining image classification models. We also thank B. O'Donoghue for useful remarks about the manuscript, and L. Schmidt for an in-depth discussion of their results on this topic. Finally, we thank D. Balduzzi, S. Legg, K. Kavukcuoglu and J. Martens for encouragement, support, lively discussions and feedback.

## Footnotes

[1]Throughout the paper, we use the words "example" and "point" interchangeably.

[2]Note that the adversarial error estimator's goal is to estimate the error rate, not the adversarial error rate (i.e., the error rate on the adversarial examples).

[3] For an event $B$, $\mathbb{I}(B)$ denotes its indicator function: $\mathbb{I}(B) = 1$ if $B$ happens and $\mathbb{I}(B) = 0$ otherwise.

[4]Note that this assumption limits the applicability of our method, excluding such centered or essentially centered image classification benchmarks as MNIST [20] or CIFAR-10 [18].

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
