[Supplementary Material]

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

## A  Synthetic experiments

In this section, we present full details of the experiments on a simple synthetic classification problem, which we presented briefly in Section 4. These experiments illustrate the power of the method of Section 3. The advantage of the simple setup considered here is that we are able to compute the density $h_g$ in an analytic form (see Figure 6 for an illustration).

### A.1  Data distribution and model

Let $\mathcal{X} = \mathbb{R}^{500}$ and consider an input distribution with a density $\rho$ that is an equally weighted mixture of two 500-dimensional isotropic truncated Gaussian distributions $N_\pm^{\text{trunc}}(\mu_\pm, \sigma^2 I)$ with coordinate-wise standard deviation $\sigma = \sqrt{500}$ ($I$ denotes the identity matrix of size $500 \times 500$), means $\mu_\pm = [\pm 1, 0, 0, \ldots, 0]$ and densities $\rho_\pm$ truncated in the first dimension such that $\rho_+(x) = 0$ if $x_1 \leq 0.025$ and $\rho_-(x) = 0$ if $x_1 \geq -0.025$. The label of an input point $x$ is $f^*(x) = \text{sgn}(x_1)$, which is the sign of its first coordinate.

We consider linear classifiers of the form $f(x) = \text{sgn}(w^\top x + b)$ trained with the cross-entropy loss $c((w, b), x, y) = \ln(1 + e^{-y(w^\top x + b)})$ where $y = f^*(x)$. We employ a one-step gradient method (which is an $L_2$ version of the fast gradient-sign method of [14, 23]) to define our AEG $g$, which tries to modify a correctly classified point $x$ with label $y$ in the direction of the gradient of the cost function $c$: $x' = x + \varepsilon \nabla_x c((w, b), x, y)/\|\nabla_x c((w, b), x, y)\|_2$ for some $\varepsilon > 0$. For our specific choice of $c$, the above simplifies to $x' = x - \varepsilon y w/\|w\|_2$. To comply with the requirements for an AEG, we define $g$ as follows: $g(x) = x'$ if $L(f, x) = 0$ and $f^*(x) = f^*(x')$ (corresponding to (G2) and (G1), respectively), while $g(x) = x$ otherwise. Therefore, if $x'$ is misclassified by $f$, $x$

Figure 6: Illustration of the data distribution and the linear model $f(x) = \text{sgn}(w^\top x + b)$ in two dimensions. The blue and green gradients show the probability density $\rho$ of the data with true labels $y = -1$ and $y = 1$, respectively, while the white space between them is the margin with $\rho = 0$. The red line is the model's classification boundary with its parameter vector $w$ shown by the purple arrow. Depending on the label, $w$ or $-w$ is the direction of translation used to perturbed the correctly classified data points, and the translations used by the AEG $g$ for specific points are depicted by grey arrows: solid arrows indicate the cases where $g(x) = x' \neq x$, while dashed arrows are for candidate translations which are not performed by the AEG because they would change the true label, $f^*(x') \neq f^*(x)$, and hence $g(x) = x \neq x'$. Each original/perturbed data point is represented by a color-coded circle: the inner color corresponds to the true label (dark blue for $y = -1$ and dark green for $y = 1$) while the outer color to the model's prediction (dark blue for $f(x) = -1$ and dark green for $f(x) = 1$). Points $x'_A$ and $x'_B$ can be obtained from $x_A$ and $x_B$, respectively, by applying the AEG, $x'_A = g(x_A)$ and $x'_B = g(x_B)$. Since only $x_A$ is mapped to $x'_A$ by $g$, and $g(x'_A) = x'_A$, the density (Radon-Nikodym derivative) can be obtained as $h_g(x'_A) = \rho(x'_A)/(\rho(x_A) + \rho(x'_A)) \in (0, 1)$. In the case of $x'_B$, $h_g(x'_B) = \rho(x'_B)/(\rho(x_B) + \rho(x'_B)) = 0$ due to the margin. Note that the formula for $h_g(x'_A)$ does not depend on whether $x_A$ or $x'_A$ is in the original test set $S$; in the first case we call $x'_A$ a "successful adversarial example" while in the second case $x'_A$ is called "originally misclassified" (a similar argument holds for $h_g(x'_B)$). $x'_C$ is not a successful adversarial example since $L(f, x'_C) = 0$ (however, $g(x_C) = x'_C$ according to our definition). Points $x_D$ and $x_E$ are not perturbed by our AEG, since $f^*(x_D) \neq f^*(x'_D)$ and $f^*(x_E) \neq f^*(x'_E)$.

and $x'$ are the only points mapped to $x'$ by $g$. Thus, the density at $x'$ after the transformation $g$ is $\rho'(x') = \rho(x) + \rho(x')(1 - L(f, x))\mathbb{I}(f^*(x) = f^*(x'))$ and

$$h_g(x') = \frac{\rho(x')}{\rho'(x')} = \frac{\rho(x')}{\rho(x') + \rho(x)(1 - L(f, x))\mathbb{I}(f^*(x) = f^*(x'))}$$

(note that $\mathbb{I}(L(f, x) = 0) = 1 - L(f, x)$).

## A.2 Experiment setup

We present two experiments showing the behavior of our independence test: one where the training and test sets are independent, and another where they are not.

In the first experiment a linear classifier was trained on a training set $S_{\text{Tr}}$ of size 500 for 50,000 steps using the RMSProp optimizer [29] with batch size 100 and learning rate 0.01, obtaining zero (up to numerical precision) final training loss $c$ and, consequently, 100% prediction accuracy on the training data. Then the trained classifier was tested on a large test set $S_{\text{Te}}$ of size 10,000.[5] Both sets were drawn independently from $\rho$ defined above. We used a range of $\varepsilon$ values matched to the scale of the data distribution: from $10^{-2}$, which is the order of magnitude of the margin between two classes (0.05), to $10^2$, which is the order of magnitude of the width of the Gaussian distribution used for each classes ($\sigma = \sqrt{500}$).

In the second experiment we consider the situation where the training and test sets are not independent. To enhance the effects of this dependence, the setup was modified to make the training process more amenable to overfitting by simulating a situation when the model has a wrong bias (this may happen in practice if a wrong architecture or data preprocessing method is chosen, which, despite the modeler's best intentions, worsens the performance). Specifically, during training we added a penalty term $10^4 w_1^2$ to the training loss $c$, decreased the size of the test set to 1000 and used 50% of the test data for training (the final penalized training loss was 0.25 with 100% prediction accuracy on the training set). Note that the small training set and the large penalty on $w_1$ yield classifiers that are essentially independent of the only interesting feature $x_1$ (recall that the true label of a point $x$ is $\text{sgn}(x_1)$) and overfit to the noise in the data, resulting in a true model risk $R(f) \approx 1/2$.

## A.3 Results

The results of the two experiments are shown in Figure 7, plotted against different perturbation strengths: the left column corresponds to the first experiment while the right column to the second. The first row presents the $p$-values for rejecting the independence hypothesis, calculated by repeating the experiment (sampling data and training the classifier) 100 times and applying the single-model (Section 3, labelled as $N = 1$ in the plots) and $N$-model (Section 3.3, labelled as $N = 2, 10, 25, 100$ in the plots) independence test, and taking the average over models (or model sets of size $N$) for each $\varepsilon$. We also plot empirical 95% two-sided confidence intervals ($N \leq 2$) or, due to limited number of $p$-values available after dividing 100 runs into disjoint bins of size $N \geq 10$, ranges between minimum and maximum value ($N = 10, 25$). For all methods of detecting dependence, it can be seen that for the independent case the test is correctly not able to reject the independence hypothesis (the average $p$-value is very close to 1, although in some runs it can drop to as low as 0.5). On the other hand, for $10 \leq \varepsilon \leq 50$, the non-independent model failed the independence test at confidence level $1 - \delta \approx 100\%$, hence, in this range of $\varepsilon$ our independence test reliably detects overfitting.

In fact, it is easy to argue that our test should only work for a limited range of $\varepsilon$, that is, it should not reject independence for too small or too large values of $\varepsilon$. First we consider the case of small $\varepsilon$ values. Notice that except for points $g(x)$ $\varepsilon$-close (in $L_2$-norm) to the true decision boundary or the decision boundary of $f$, $g(x)$ is invertible: if $g(x)$ is correctly classified and is $\varepsilon$-away from the true decision boundary, there is exactly one point, $x$, which is translated to $g(x)$, while if $g(x)$ is incorrectly classified and $\varepsilon$-away from the decision boundary of $f$, no translation leads to $g(x)$ and $x = g(x)$; any other points are $\varepsilon$-close to the decision boundary of either $f$ or $f^*$. Thus, since $\rho$ is bounded, $g(x)$ is invertible on a set of at least $1 - O(\varepsilon)$ probability (according to $\rho$). When $\varepsilon \to 0$, $g(x) \to x$, and so $\rho(g(x)) \to \rho(x)$ for all points $x$ with $|x_1| \neq 0.025$ (since $\rho$ is continuous in all such $x$), implying $h_g(g(x)) \approx 1$ on these points. It also follows that $L(f, x) \neq L(f, g(x))$ can only

Figure 7: Risk and overfitting metrics for a synthetic problem with linear classifiers as a function of the perturbation strengths $\varepsilon$ (log scale). Left: unbiased model tested on a large, independent test set (in this case $\widehat{R}_S(f) \approx \widehat{R}_g(f) \approx R(f)$); right: trained model overfitted to the test set ($\widehat{R}_S(f) \leq \widehat{R}_g(f)$ while both are smaller than $R(f)$). *First row*: Average $p$-value $\delta$ for the pairwise independence test with over 100 runs ($N = 1$) or the $N$-model independence test ($N > 1$). The bounds plotted are either empirical 95% two-sided confidence intervals ($N \leq 2$) or ranges between minimum and maximum value ($N = 10, 25$). *Second row*: Empirical two-sided 97.5% confidence intervals for the empirical test error rate $\widehat{R}_S(f)$ and the adversarial risk estimate $\widehat{R}_g(f)$. On the left, $R(f) \approx \widehat{R}_S(f)$, while $R(f)$ is shown separately on the right. *Third row*: Average densities (Radon-Nikodym derivatives) for originally misclassified points and for the new data points obtained by successful adversarial transformations (with empirical 97.5% two-sided confidence intervals). *Fourth row*: The empirical test error rate $\widehat{R}_S(f)$ and the adversarial risk estimate $\widehat{R}_g(f)$ for a single realization with 97.5% two-sided confidence intervals computed from Bernstein's inequality, the adversarial error rate $\widehat{R}_{S'}(f)$, and the expected error $R(f)$ (on the right, on the left $R(f) \approx \widehat{R}_S(f)$). *Fifth row*: Histograms of $p$-values for selected $\varepsilon$ values over 100 runs.

happen to a set of points with an $O(\varepsilon)$ $\rho$-probability. This means that $L(f, g(x))h_g(g(x)) \approx L(f, x)$ on a set of $1 - O(\varepsilon)$ $\rho$-probability, and for these points $T_g(x) = L(f, g(x))h_g(g(x)) - L(f, x) \approx 0$. Thus, $T_g(X) \approx 0$ with $\rho$-probability $1 - O(\varepsilon)$. Unless the test set $S$ is concentrated in large part on the set of remaining points with $O(\varepsilon)$ $\rho$-probability, the test statistic $|T_{S,g}(f)| = O(\varepsilon)$ with high probability and our method will not reject the independence hypothesis for $\varepsilon \to 0$.

When $\varepsilon$ is large ($\varepsilon \to \infty$), notice that for any point $x$ with non-vanishing probability (i.e., with $\rho(x) > c$ for some $c > 0$), if $g(x) \neq x$ than $\rho(g(x)) \approx 0$. Therefore, for such an $x$, if $L(f, x) = 0$ and $L(f, g(x)) = 1$, $h_g(g(x)) = \rho(g(x))/(\rho(x) + \rho(g(x))) \approx 0$, and so $T_g(x) \approx 0$ (if $L(f, g(x)) = 0$, we trivially have $T_g(x) = 0$). If $L(f, x) = 1$, we have $g(x) = x$. If $g$ is invertible at $x$ then $h_g(x) = 1$ and $T_g(x) = 0$. If $g$ is not invertible, then there is another $x'$ such that $g(x') = x$; however, if $\rho(x) > c$ then $\rho(x') \approx 0$ (since $\varepsilon$ is large), and so $h_g(g(x)) = \rho(x)/(\rho(x) + \rho(x')) \approx 1$, giving $T_g(x) \approx 0$. Therefore, for large $\varepsilon$, $T_g(X) \approx 0$ with high probability (i.e., for points with $\rho(x) > c$), so the independence hypothesis will not be rejected with high probability.

To better understand the behavior of the test, the second row of Figure 7 shows the empirical test error rate $\widehat{R}_S(f)$, the (unadjusted) adversarial error rate $\widehat{R}_{S'}(f)$, and the adversarial risk estimate $\widehat{R}_g(f)$, together with their confidence intervals. For the non-independent model, we also show the expected error $R(f)$ (estimated over a large independent test set), while it is omitted for the independent model where it approximately coincides with both $\widehat{R}_S(f)$ and $\widehat{R}_g(f)$. While the reweighted adversarial error estimate $\widehat{R}_g(f)$ remains the same for all perturbations in case of an independent test set (left column), the adversarial error rate $\widehat{R}_{S'}(f)$ varies a lot for both the dependent and independent test sets. For example, in the case when the test samples and the model $f$ are not independent, it undershoots the true error for $\varepsilon < 10$ and overshoots it for larger perturbations. For very large perturbations ($\varepsilon$ close to 100), the behavior of $\widehat{R}_{S'}(f)$ depends on the model $f$: in the independent case $\widehat{R}_{S'}(f)$ decreases back to $\widehat{R}_S(f)$ because such large perturbations increasingly often change the true label of the original example, so less and less adversarial points are generated. In the case when the data and the model are not independent (right column), the adversarial perturbations are almost always successful (i.e., lead to a valid adversarial example for most originally correctly classified points), yielding an adversarial error rate close to one for large enough perturbations. This is because the decision boundary of $f$ is almost orthogonal to the true decision boundary, and so the adversarial perturbations are parallel with the true boundary, almost never changing the true label of a point.

The plots of the densities (Radon-Nikodym derivatives), given in the third row of Figure 7, show how the change in their values compensate the increase of the adversarial error rate $\widehat{R}_{S'}(f)$: in the independent case, the effect is completely eliminated yielding an unbiased adversarial error estimate $\widehat{R}_g(f)$, which is essentially constant over the whole range of $\varepsilon$ (as shown in the first row), while in the non-independent case the similar densities do not bring back the adversarial error rate $\widehat{R}_{S'}(f)$ to the test error rate $\widehat{R}_S(f)$, allowing the test to detect overfitting. Note that the densities exhibit similar trends (and values) in both cases, driven by the dependence of typical values of the $\rho(x)/\rho(g(x))$ ratio on the perturbation strength $\varepsilon$ for originally misclassifed points ($L(f, x) = 1$) and for successful adversarial examples (i.e., $L(f, x) = 0$ and $L(f, g(x)) = 1$).

To compare the behavior of our improved, pairwise test and the basic version, the fourth row of Figure 7 depicts a single realization of the experiments where the $97.5\%$ confidence intervals (as computed from Bernstein's inequality) are shown for the estimates. For the independent case, the confidence intervals of $\widehat{R}_S(f)$ and $\widehat{R}_g(f)$ overlap for all $\varepsilon$, and thus the basic test is not able to detect overfitting. In the non-independent case, the confidence intervals overlap for $\varepsilon = 10$ and $\varepsilon = 75$, thus the basic test is not able to detect overfitting with at a $95\%$ confidence level, while the improved test (second row) is able to reject the independence hypothesis for these $\varepsilon$ values at the same confidence level.

Finally, in the fifth row of Figure 7 we plotted the histograms of the empirical distribution of $p$-values for both models, over 100 independent runs (between the runs, all the data was regenerated and the models were retrained). For $\varepsilon = 0.1, 5, 20$, they concentrate heavily on either $\delta = 0$ or $\delta = 1$, and have very thin tails extending far towards the opposite end of the $[0, 1]$ interval. This explains the surprisingly wide $95\%$ confidence intervals for $p$-values plotted in the first row. In particular, the fact that some $p$-values for the independent model are as low as $0.5$ does not mean the independence test is not reliable, because almost all calculated $\delta$ values are close or equal to 1, and the few outliers are

Figure 8: Histograms of $p$-values from $N$-model ($N = 1, 2, 10$) independence tests for both synthetic models and selected $\varepsilon$ values, over 100 runs.

a combined consequence of the finite sample size and the effectiveness of the AEG. The additional $\varepsilon = 6$ histogram for the non-independent model illustrates a regime which is in between the single-model pairwise test (Section 3) completely failing to reject the independence hypothesis and clearly rejecting it.

To verify experimentally whether the $N$-model independence test can be a more powerful detector of overfitting than the single-model version, in Figure 8 (right panel) we plotted $p$-value histograms for $N = 1, 2, 10$ for the intermediate AEG strength $\varepsilon = 6$ applied to the non-independent model over 100 training runs. Indeed, as $N$ increases, the concentration of $p$-values around in the low ($\delta \leq 0.2$) range increases. For $N > 10$ we did not have enough values to plot a histogram: for $N = 25$ we obtained $\delta = 0.1851, 0.1599, 0.0661$ and $0.1941$, while for $N = 100$ the $p$-value is $0.1153$. The increase of the test power becomes apparent when we compare the last value with the mean of $p$-values obtained by testing every training run separately, equal $0.5984$, and the median $0.6385$.

For comparison, we also plotted in Figure 8 (left panel) the corresponding histograms for the independent model and a slightly higher attack strength, $\varepsilon = 10$, at which the independence tests fails for the overfitted model even without averaging (see Figure 7, first row, right panel). The histograms are all clustered in the $\delta$ region close to 1, indicating that the $N$-model test is not overly pessimistic.

## B  Translational AEGs for image classification models

For image classification we consider two translation variants that are used in constructing a translational AEG. For every correctly classified image $x$, we consider translations from $\mathcal{V}_\varepsilon$ (for some $\varepsilon$), choosing $g(x)$ from the set $G(x) = \{\tau_v(x) : v \in \mathcal{V}_\varepsilon\} \cup \{x\}$. If all translations result in correctly classified examples, we set $g(x) = x$. Otherwise, we use one of two possible ways to select $g(x)$ (and we call the resulting points successful adversarial examples):

- *Strongest perturbation:* Assuming the number of classes is $K$, let $l(f, x) \in \mathbb{R}^K$ denote the vector of the $K$ class logits calculated by the model $f$ for image $x$, and let $l_{\text{exc}}(f, x) = \max_{0 \leq i < K} l_i(f, x) - l_y(f, x)$. We define

$$g_{\text{strongest}}(x) = \text{argmax}_{x' \in G(x)} \, l_{\text{exc}}(f, x'),$$

with ties broken deterministically by choosing the first translation from the candidate set, going top to bottom and left to right in row-major order. Thus, here we seek a non-identical "neighbor" that causes the classifier to err the most, reachable from $x$ by translations within a maximum range $\varepsilon$.

- *Nearest misclassified neighbor:* Here we aim to find the nearest image in $G(x)$ that is misclassified. That is, letting $d(x, x') = \|v\|_2$ if $x' = \tau_v(x)$ and $\infty$ otherwise, we define

$$g_{\text{nearest}}(x) := \text{argmin}_{x' \in G(x), L(f, x')=1} \, d(x, x')$$

with ties broken deterministically as above.

The two perturbation variants are successful on exactly the same set of images, hence they lead to the same adversarial error rates $\widehat{R}_{S'}(f)$. However, they are characterized by different values of the density $h_g$ and, consequently, yield different adversarial risk estimates $\widehat{R}_g(f)$ and associated $p$-values

for the independence test. The main difference between them is that the "strongest" version is more likely to map multiple images to the same adversarial example, thus decreasing the densities for successful adversarial examples and, counterintuitively, increasing them for originally misclassified points (as their neighbors are less likely to be mapped to these points).

To better see the effect of adversarial perturbations, we also consider two random baselines that do not take into account the success of a translation in generating misclassified points: $g_{\mathrm{random}}(x)$ is chosen uniformly at random from $G(x) \setminus \{x\}$, and $g_{\mathrm{random2}}(x)$ is chosen uniformly at random from $G(x)$.

## B.1  Maximum translations

In practice, translating an image is not always simple, as the new image has to be padded with new pixels. When (central) crops of a larger image are used (as is typical for ImageNet classifiers), translations can easily be implemented as long as the resulting new cropping window stays within the original image boundaries. Even if an image can be translated by a vector $v$, this limits our ability to compute $h_g(x')$ for the adversarial image $x'$ by (8) or (9) for $g_{\mathrm{strongest}}$ or $g_{\mathrm{nearest}}$. Indeed, if an image $x$ is shifted by $v \in \mathcal{V}_\varepsilon$ to generate adversarial example $x'$, we need to examine translations of $x'$ with vectors in $\mathcal{V}_\varepsilon$ to find the neighbors $x''$ of $x'$ potentially contributing to $n(x')$ when computing $h_g(x')$. Finally we need to consider translations of $x''$ with vectors in $\mathcal{V}_\varepsilon$ to determine the exact value they contribute, that is, to compute the exact probabilities in (9) (see Figure 9 for an illustration). Thus, to be able to compute the density $h_g$ for the adversarial points obtained by translations from $\mathcal{V}_\varepsilon$, we might need to be able to perform translations within $\mathcal{V}_{3\varepsilon}$.

Figure 9: Image translations which need to be considered for a translational AEG with $\varepsilon = 3$. The red, blue and green balls represent the center of the original image $x$, adversarial example $x' = g(x)$ and another image $x''$ contributing to $\rho_g(x')$, respectively, while the semi-translucent squares of corresponding colors represent the possible translations which need to be considered for each of $x$, $x'$ and $x''$. Solid light grey arrows represent the relationships $x' = g(x)$ and $x' = g(x'')$. Finally, the dashed arrow and the semi-translucent grey ball represent an alternative mapping, which has to be ruled out while calculating the value of $g(x'')$ and, consequently, of $h_g(x')$. It is easy to see that the colored squares (which contain the translations needing to be evaluated) extend as far as $3\varepsilon$ from the original image $x$.