[Reviews · NeurIPS 2019]

Reviewer 1



The work addresses the issue of neural networks’ overfitting to test sets on classification tasks due to widespread reuse of the same datasets throughout the community, and how that affects the credibility of reported test error rates, which should reflect performance on ‘truly new’ data from the same distribution. The proposed test statistic does not affect the training procedure, and is simple in theory: if the (importance-reweighted) empirical risk and the empirical risk of adversarially-perturbed examples differs by more than a certain threshold (given by concentration bounds), the null hypothesis that the classifier and the test data are independent is rejected. My main concern is that the type of adversarial examples used, bounded translational shifts (for image data), is very limited and likely to be unrealistic. Effectively shifting the frame of a CIFAR image is quite different from swapping items in a scene; it is less subtle and less ‘insidious’, unless perhaps a “7” is converted via truncation into a “1”. It would have been nice to see example adversarial images for a sense of how they compare to the ones typically discussed in the literature, particularly as a selling point of the work is the use of adversarial examples. Finally, while the writing is fluid, it comes across as rather too conversational and verbose. It would have been nice to have the math separated out from the text, in the form of definitions, propositions, etc., and for the text to have been generally more terse. *** [UPDATE] It appears that the authors misunderstood my examples of adversarial examples. By "swapping items in a scene", I meant repositioning or replacing irrelevant items; this is comparable to [1], which studied the addition of irrelevant items (eg, stickers to a stop sign, which happens often enough in real life, if not in the exact form shown in the paper). This was meant to be an example of a realistic adversarial example--and one not achievable by translational attacks. By "truncation of a '7' into a '1'", I meant an example that *could* be achieved by the authors' translational approach, but which is not subtle enough to really be considered an adversarial example in the classic sense. Conversely, as the authors concede, translations that fall short of such an extreme are rather weaker than standard ones. These include pixel-by-pixel perturbation (ie, adding a random mask to an image), one of the most widely studied cases, and where differences are essentially undetectable to human observers. This is a very intriguing line of work, but unless more compelling sorts of adversarial examples are used (finding ways of dealing with the induced distribution shift, as mentioned), it seems a bit premature. Hence my score remains the same. [1] Physical Adversarial Examples for Object Detectors Eykholt et al., 2018

Reviewer 2



This paper examines how overfitting in deep-learning classification can be detected through the generation of adversarial examples from the original test data. The authors have developed an unbiased estimator of error based on adversarial examples that relies on a form of importance weighting. This estimate of error should be in agreement with estimates based on the training data - otherwise, if hypothesis testing shows that the discrepancy is sufficiently great, we can conclude that the learner has overfit to the training data. The authors take great care to avoid the trap of other uses of data perturbation, where the generated data may follow distributions that do not reflect the original data. They outline a generation process that uses certain density preservation assumptions to ensure unbiased estimation of the error of the original classification process, and then give practical ways of generating adversarial examples for images that are valid for these assumptions. The authors then illustrate their method in two learning scenarios (involving CIFAR-10 and ImageNet). The authors are also careful in addressing the practical issues surrounding the use of their estimator, including that of managing variation in the estimation outcome. Overall, this is a good result, well motivated, well written and well explained. [UPDATE] The issues raised by the first author regarding the generation of adversarial examples by translation alone seems neither here nor there to me. Any adversarial perturbations used should remain realistic, and translations are a good way of ensuring this. I am satisfied with the authors' response, and my score is unchanged.

Reviewer 3



This paper proposes a novel way to detect overfitting via adversarial examples. The authors construct a new unbiased (error) estimator which uses adversarial examples as well as importance weighting to take into account the change in data distribution. Moreover, the authors use translation attack in this paper which simplifies the computation of importance weighting. This method is quite general as it doesn't restrict the training procedure and is based on the original test data. The theoretical analysis for the high probability error bound given independence assumption looks correct. The paper provides a thorough experimental validation of the proposed algorithm, showing that there's strong evidence that the current state-of-the-art classifier architecture are overfitted to the CIFAR10, and no such evidence in case of ImageNet. I like this work a lot as it makes progress towards the answering an important question to deep learning community --- whether the recent progress on model architectures come from overfitting the benchmark test dataset? Different from the reason given by Recht et al.(distribution shift), the extensive experiments of this paper suggests that the test error on CIFAR10 are unreliable and due to overfitting. Also, this work is well-written and structured clearly. Thus I suggest accepting this work. =========post-rebuttal update======== The authors clarifies my questions in rebuttal and my score remains the same.

[Author Response · NeurIPS 2019]

We thank the Reviewers for their thoughtful assessment of our work and valuable comments. Below we address the main questions raised in the reviews.

**Reviewer 1**

- *About the employed adversarial attacks:* Adversarial attacks are usually meant to be subtle, making unnoticeable changes in the original image (and keeping the original label unchanged). Thus, "swapping items in a scene" or converting a 7 via truncation into a 1, as mentioned in the review, are usually not considered to be adversarial attacks. Indeed, translation attacks (introduced by Azulay and Weiss, 2018) are much less effective than the most popular gradient-based attacks (e.g., the fast gradient sign method or PGD), however, the effects of the latter are often noticeable on the images (while keeping the labels intact) and change the image distribution quite a lot. Our original intention was to use the gradient-based attacks, however, we could not deal properly with the resulting distribution shift (i.e., calculate the corresponding Radon-Nikodym derivative). Translation attacks were selected because there we could address this issue, and—perhaps surprisingly—they work reasonably effectively (on CIFAR-10, with a standard test set error rate of about 2-8%, the adversarial translations were successful for an additional 5-8% of the test set, while on ImageNet, with the standard test set error rates in the range of 20-30%, adversarial attacks were successful for an additional 7-13% of the test images). We also think that translation attacks are quite realistic in the sense that they capture the real-life phenomenon that photos of the same subjects can be framed differently, for example when trying out different compositions or taking several photos in a sequence. From this point of view, image translations are actually more realistic than adding model-dependent noise, which is essentially what gradient-based attacks do. Figure 1 on page 2 shows an attack example for ImageNet; we will include more examples in the appendix, also for CIFAR-10.

- We will work on improving the writing for the final version, as suggested. We will aim to make the text more concise (without making it harder to follow) in the camera-ready version, separate out the mathematical formalism and include more adversarial examples in the supplementary material. We will add a more complete justification for translational attacks, either in the main text or as an appendix.

- Just for clarity, we would like to point out that our method was also applicable to ImageNet, where we found overfitting to the training set but not to the test set.

**Reviewer 2**

The test can naturally be applied at any point of the training process to see if overfitting has happened. However, our independence test itself uses the test data, so using the results of this test already leaks information from the test set to the training process, hence induces some degree of overfitting. Also, using the test multiple times increases the risk of a false positive, which one has to protect against by using, e.g., the Bonferroni correction (i.e., applying a union bound over the Type I error probabilities of the multiple tests).

**Reviewer 3**

- We will provide more details about the experimental settings and the training methods (including the selection of hyperparameters) in the appendix. In all the training procedures, the number of epochs and the corresponding learning rate schedules were fixed in advance, following the recommendations of previous work in the literature. We used different random seeds for each training process.

- Indeed, hyperparameter selection is one of the potential sources of overfitting. When averaging over multiple i.i.d. training runs (as we do in our strongest tests), the only possible causes of overfitting are tweaking either (i) the hyperparameters or (ii) the model architecture in order to minimize the test set error rate. The suggestion of tuning the hyperparameters to CIFAR-10.1 could help to distinguish between the two: if choosing the hyperparameters of a CIFAR-10 model (trained on a CIFAR-10 training set) to minimize the CIFAR-10.1 test set error rate were to lead to a model overfitted to the latter but not to the CIFAR-10 test set, it would suggest that (i) is a more important source of overfitting than (ii).

[Meta-Review · NeurIPS 2019]

The reviewers all liked the results in the paper on gauging overfitting via adversarial samples (particularly given that benchmark datasets are widely reused). Reviewers have suggested modifications of possibly better adversarial samples, and additional information on the experimental settings.